# A Smooth Muscle Cell-Based Ferroptosis Model to Evaluate Iron-Chelating Molecules for Cardiovascular Disease Treatment

Sarah El Hajj [1,2], Laetitia Canabady-Rochelle [2], Isabelle Fries-Raeth [2] and Caroline Gaucher [1,*]

1   Université de Lorraine, CITHEFOR, F-54505 Vandoeuvre Les Nancy, France; elhajjsarah@gmail.com
2   Université de Lorraine, LRGP, CNRS, F-54000 Nancy, France;
    laetitia.canabady-rochelle@univ-lorraine.fr (L.C.-R.); isabelle.raeth@univ-lorraine.fr (I.F.-R.)
*   Correspondence: caroline.gaucher@univ-lorraine.fr

**Abstract:** Dysregulation of iron homeostasis causes iron-mediated cell death, recently described as ferroptosis. Ferroptosis is reported in many chronic diseases, such as hepatic cancer, renal, and cardiovascular diseases (heart failure, atherosclerosis). However, there is a notable scarcity of research studies in the existing literature that explore treatments capable of preventing ferroptosis. Additionally, as far as the author is aware, there is currently no established model for studying ferroptosis within cardiovascular cells, which would be essential for assessing metal-chelating molecules with the potential ability to inhibit ferroptosis and their application in the treatment of cardiovascular diseases. In this study, a smooth muscle cell-based ferroptosis model is developed upon the inhibition of the system $X_c^-$ transporter by erastin associated or not with Fe(III) overload, and its rescue upon the introduction of well-known iron chelators, deferoxamine and deferiprone. We showed that erastin alone decreased the intracellular concentration of glutathione (GSH) without affecting peroxidized lipid concentrations. Erastin with ferric citrate was able to decrease intracellular GSH and induce lipid peroxidation after overnight incubation. Only deferiprone was able to rescue the cells from ferroptosis by decreasing lipid peroxidation via iron ion chelation in a 3:1 molar ratio.

**Keywords:** ferroptosis; human aortic smooth muscle cells; intracellular GSH; lipid peroxidation; iron chelation



## 1. Introduction

Iron, an essential mineral in the human body, plays a crucial role in various physiological processes, such as hemoglobin production, oxygen transport, and enzyme function [1]. As free iron might be toxic due to its redox potential, the body stores excess iron in the liver, where it can be mobilized when needed [2]. Iron storage is regulated to maintain a balance between absorption and utilization [3]. Maintaining iron homeostasis is critically essential for human health to prevent diseases such as iron deficiency anemia and hemochromatosis caused by iron overload [4]. The concentration of intracellular iron ions is regulated at several cellular levels that comprise iron ($Fe^{3+}$) uptake via transferrin receptors (TFR1 and TFR2), iron ($Fe^{2+}$) storage in ferritin, and iron ($Fe^{2+}$) export via ferroportin [5]. In the United States and France, the Recommended Dietary Allowance is 8 mg or 9 mg of iron for adult males and 18 mg or 16 mg for adult females to compensate for the losses during menstrual bleeding increasing to 27 mg or 35 mg during pregnancy, respectively [6]. Moreover, the Recommended Dietary Allowance for vegetarians is 1.8 times higher than for people who eat meat. This intake will meet the daily needs of 1 mg for adult males and 2 mg for adult females due to intestinal absorption limited at 5 to 10% [7]. Dysregulation of iron homeostasis causes iron-mediated cell death, recently described as ferroptosis, which is genetically, biochemically, and morphologically different from apoptosis and necrosis [8,9]. Indeed, free iron ions catalyze the formation of hydroxyl radicals via the Haber–Weiss

reaction [10] or can be a direct reactant for their production, such as Fe(II) in the Fenton reaction [11].

Ferroptosis is a programmed cell death implicated in the development of several diseases, such as hepatic cancer, renal, and cardiovascular diseases (heart failure, atherosclerosis) [12]. The hallmark feature of ferroptosis is iron-dependent accumulation of oxidized phospholipids (i.e., lipid peroxides) [13]. Ferroptosis can be "mimicked"/induced by the inactivation of system $X_c^-$ (cystine/glutamate antiporter) using either erastin, sorafenib, sulfasalazine, or excess glutamate or also by inhibiting glutathione peroxidase 4 using Ras Selective Lethal 3. The inhibition of the system $X_c^-$ induces the depletion of the intracellular cysteine pool, which is the limiting amino acid required for GSH synthesis [14,15]. Yet, the decrease in intracellular GSH concentration causes an imbalance in cell redox homeostasis and antioxidant capacity. Indeed, it inhibits glutathione-dependent enzymes, such as the glutathione peroxidase-4, that catalyze the reduction of lipid peroxides, while oxidizing GSH, eventually helping protect cells from oxidative stress-induced plasma membrane injury [15–17].

Ferroptosis can be prevented by inhibiting lipid peroxidation using either the natural vitamin E, the synthetic ferrostatin-1 and liproxstatine-1, or by chelating iron ions ($Fe^{2+/3+}$) using deferoxamine (DFO) or deferiprone (DFr) [15]. Although ferroptosis has been identified as a cell death pathway occurring in cardiovascular diseases, such as atherosclerosis, very few studies in the literature report treatments inhibiting ferroptosis [18,19]. Moreover, to the author's knowledge, there is no model of ferroptosis based on cardiovascular cells in order to test molecules that are able to inhibit ferroptosis with potential application in cardiovascular disease treatment.

Hence, the objective of this study is to develop a cell-based ferroptosis model upon the inhibition of the system $X_c^-$ antiporter and triggering lipid peroxidation with Fe(III) overload. This model will be validated using Fe(III) chelators, such as DFO and Dfr, as ferroptosis rescue molecules.

## 2. Materials and Methods

### 2.1. Reagents

Erastin (E7781, Sigma–Aldrich, Saint-Quentin-Fallavier, France) mother solution was prepared at an initial concentration of 10 mM in 100% DMSO (FLUKA), aliquoted and stored at −80 °C. Ferric citrate (F3388-Sigma–Aldrich, Saint-Quentin-Fallavier, France) mother solution was prepared at an initial concentration of 25 mM in warm PBS (37 °C) and stored at 4 °C. Deferoxamine (D0160000, European Pharmacopia, Starsbourg, France) mother solution was prepared at an initial concentration of 30 mM in PBS, aliquoted, and stored at −80 °C. Deferiprone (Y0001976, European Pharmacopia, Starsbourg, France) mother solution was prepared at an initial concentration of 134 mM in warm ultra-pure water, aliquoted, and stored at −80 °C.

### 2.2. Cell Culture

Smooth muscle cells derived from human aorta, AoSMC (CC-2571, LONZA, Colmar, France) were grown (37 °C, 10% $CO_2$) in smooth muscle complete cell growth medium (CC-3182, LONZA, Colmar, France) containing SmBM$^{TM}$ Basal Medium and SmGM$^{TM}$ 2 SingleQuots supplements. Supplements comprise 5% (*v/v*) of fetal bovine serum (FBS), 0.1% (*v/v*) insulin, 0.2% (*v/v*) of human basic fibroblast growth factor (hFGF-B), 0.1% (*v/v*) of gentamicin, 0.1% (*v/v*) of amphotericin-B, and 0.1% (*v/v*) of human epidermal growth factor (hEGF). All experiments were conducted using actively growing cells at around 90% of confluency.

### 2.3. Cytocompatibility Test

The cytocompatibility of erastin at a concentration range comprised between 0.01 μM and 10 μM was evaluated using the MTT test. AoSMC were seeded at a density of $60 \times 10^3$ cells/mL of complete growth medium in a 96-well plate. After 24 h, cells were

incubated with a different concentration of erastin for another 24 h at 37 °C. Then, 50 µL of 3-(4,5-dimethylthiazol-2-yl)-2,5-diphenyltetrazolium bromide (MTT) solution (5 mg/mL) were added to each well for 3 h at 37 °C. In viable cells, cellular oxidoreductase can reduce MTT to its insoluble form, called formazan, that has a purple color. The formazan crystals were dissolved with 50 µL of 100% DMSO in each well, and the absorbance was read at a wavelength of 570 nm.

### 2.4. Induction and Rescue of Ferroptosis

AoSMCs were seeded at a density of $60 \times 10^3$ cells/mL of complete growth medium in a 6-well plate (for further GSH, MDA, and protein quantifications). Solutions of 1% DMSO (100%), 100 µM erastin, 500 µM FC, 500 µM DFr, and 1500 µM DFO solution were prepared in a warm (37 °C) incubation medium (complete cell growth medium either undiluted or diluted 2 times with PBS). Those solutions were further diluted 10 times in a final volume of 1 mL of complete growth medium to obtain final concentrations of 0.1% DMSO (for control), 10 µM erastin, 50 µM FC (for inducing cellular ferroptosis), 50 µM DFr, and 150 µM DFO (for rescuing cells from ferroptosis) and were incubated overnight (16 h, 37 °C, 10% $CO_2$) with AoSMC. According to some studies, 10 µM of erastin is the commonly used concentration to inhibit the system $X_c^-$ and induce ferroptosis [20–22]. Therefore, a concentration of 10 µM for erastin was chosen in order to ensure the inhibition of system $X_c^-$ necessary for the development of the ferroptosis model on AoSMC in our experiments.

### 2.5. Quantification of Intracellular Glutathion Concentration

After incubation, cells were lysed with 250 µL of 3.3% (*v/v*) cold perchloric acid (Sigma–Aldrich) and centrifuged at 10,000× *g* during 15 min at 4 °C. A total of 200 µL of the supernatant were neutralized by 11.5 µL NaOH-40% (*v/v*), whereas the pellet was collected and resuspended in 200 µL of lysis buffer (0.5 M Tris-HCl, 1.5 M NaCl, 10% SDS, and 10% (*v/v*) of 10% (*w/v*) Triton X-100) and frozen at −80 °C for further protein quantification. The neutralized supernatant was diluted (4 times for control condition, 2 times for the erastin condition, and 1.3 times both for the erastin + FC condition and for erastin + DC + DFO or DFr condition) in a solution containing 0.1 M HCl and 2 mM EDTA. Then, 60 µL of each previously diluted supernatant was deposited in triplicates in a 96-well black microplate. A total of 120 µL of 0.4 M borate buffer (pH 9.2; MERCK) and 20 µL of 5.4 mM 2,3-naphthalene dicarboxaldehyde (FLUKA) were then added to each well. The plate was incubated for 25 min at 4 °C. The fluorescence intensity was measured at $\lambda_{exc}$ = 485 nm and $\lambda_{em}$ = 538 nm using the microplate reader Synergy2—Bioteck and the Gen5 version 1.08 software. GSH concentration in each sample was determined from a calibration curve presenting fluorescence intensity as a function of GSH concentration ranging from 0.325 µM to 3.25 µM.

### 2.6. Quantification of Peroxidized Lipids

Malondialdehyde (MDA) was quantified as a final product of lipid peroxidation using the TBARS method [23]. After incubation, cells were lysed in 200 µL of 10% (*v/v*) perchloric acid and centrifuged at 10,000× *g* for 15 min at 4 °C. The supernatant was added with 200 µL of TBA solution (15 g of trichloroacetic acid and 0.38 g of thiobarbituric acid dissolved in 100 mL of 0.25 M HCl) and 2 µL of 2% (*w/v* in absolute ethanol) butylated hydroxytoluene, whereas the pellet was resuspended in 200 µL of lysis buffer (0.5 M Tris-HCl, 1.5 M NaCl, 10% (*v/v*) SDS, and 10% pure Triton X-100) and frozen at −80 °C for further protein quantification. The treated supernatants were incubated at 95 °C for 1 h. The reaction was stopped on ice for 3 min. The MDA-TBA adduct was then extracted with 400 µL of butanol-1 and centrifuged at 15,000× *g* for 5 min at 4 °C. A volume of 120 µL of each supernatant was deposited in triplicates in a 96-well black microplate. The fluorescence intensity was measured at $\lambda_{exc}$ = 485 nm and $\lambda_{em}$ = 538 nm using the microplate reader Jasco and the Spectra Manager version 2.10.01 software. The

concentration of MDA in each sample was determined from the plotted calibration curve representing the fluorescence intensity as a function of MDA concentration ranging from 0.0625 μM to 2.5 μM. The MDA standard (10 mM) was synthesized by diluting 208 μL of 1,1,3,3-tetramethoxypropane in 100 mL of 10% (*v/v*) perchloric acid and incubated at room temperature for 2 h in darkness. MDA concentration was calculated upon its absorbance at 245 nm (UV spectrophotometer, BioTek Microplate Reader 800TS) and its molar absorbance ($\varepsilon_{245\ nm}$ = 13,700 $mol^{-1} \cdot L \cdot cm^{-1}$) using the Beer–Lambert equation [24].

### 2.7. Quantification of Cellular Proteins

The cellular protein concentration was quantified on each pellet generated in Sections 2.5 and 2.6 to normalize the quantification of GSH and MDA, as well as to estimate a variation of cells remaining after incubations. The cellular protein concentration was determined using the bicinchoninic acid assay kit (ThermoFisher Scientific, Waltham, MA, USA), including a calibration curve of bovine serum albumin ranging from 25 μg/mL to 1000 μg/mL. A volume of 25 μL of each resuspended pellet was deposited in triplicates in a 96-well microplate and added with 200 μL working buffer. The absorbance at 570 nm was read after 30 min of incubation at 37 °C using a microplate reader (BioTeck 800TS).

### 2.8. Statistical Analysis

Results are shown as mean $\pm$ standard error of the mean (SEM); only the upper SEM limit is shown as equal to the lower one. Mean and SEM calculations are based on 3 different wells (triplicate) per group from 3 independent experiments in each group (*n* = 3). The number of samples has been reduced to the smallest size allowing statistical differences between conditions. The statistical analysis was performed using GraphPad Prism version 9 software (San Diego, CA, USA). All data were analyzed using the Shapiro–Wilks test and showed a *p*-value > 0.05, suggesting that there is no significance that the data deviates from a normal distribution. Then, comparisons were performed using a *t*-test or one-way ANOVA test followed by Tukey's post-test [25]. Differences were considered significant when *p* < 0.05.

## 3. Results

The development of the ferroptosis model on AoSMCs was conducted by first testing the cytocompatibility of erastin and, then, by following the two biomarkers: intracellular concentration of GSH and lipid peroxidation, normalized in reference to cellular protein content.

### 3.1. Cytocompatibility of Erastin

A range of erastin concentrations (0.01 μM–10 μM) was tested for evaluating their cytocompatibility on AoSMCs as a first required step in order to set the maximum concentration of erastin to be used in developing the ferroptosis model without killing the cells. For all tested concentrations of erastin, cells show a metabolic activity of approximately 80 $\pm$ 20% of the control (Figure 1). Thus, AoSMCs have maintained their functional properties and growing abilities after 24 h of treatment with erastin, whatever the investigated concentration is.

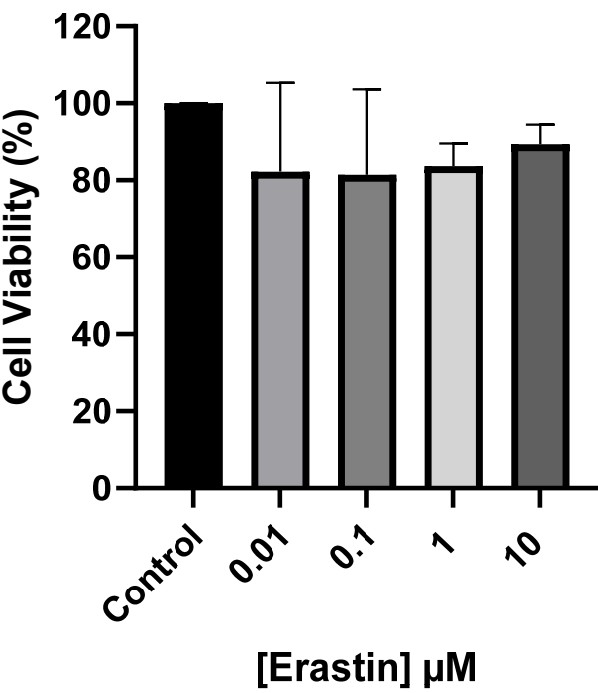

**Figure 1.** AoSMC metabolic activity after 24 h incubation with different erastin concentrations. Results are presented as mean $\pm$ sem of $n$ = 3 experiments and compared using one-way ANOVA.

*3.2. Erastin-Induced Intracellular GSH Depletion in Different Dilution of Culture Media*

To validate system $X_c^-$ inhibition, we quantified intracellular GSH concentration upon AoSMC incubation with erastin in non-diluted complete culture medium, or in 2 times or diluted complete culture medium. Indeed, cystine and glutamate present at high concentrations in culture medium (confidential information from LONZA) might have a negative impact on erastin inhibitory activity. Indeed, cystine and glutamate in excess can reach cell cytoplasm via transmembrane proteins responsible for amino acid transport such as the L-aminoacid transporter.

Figure 2 shows the intracellular concentration of GSH and proteins before and after incubation of AoSMCs with erastin (10 μM in different incubation media (diluted or not). In non-diluted complete medium and in the absence of erastin, the physiological concentration of GSH is stated at 10 $\pm$ 1.4 μM (Figure 2a) and the protein concentration at 520 $\pm$ 80 μg/mL (Figure 2b). Following erastin overnight incubation in complete culture medium, the intracellular GSH concentration (Figure 2a) decreased by 9 times compared to the control condition. The cellular protein concentration was determined to evaluate the number of cells remaining after the incubation. No significant change was observed in protein concentration between the control condition (520 $\pm$ 80 μg/mL) and the erastin condition (390 $\pm$ 20 μg/mL) (Figure 2b). In complete medium diluted 2 times, the intracellular GSH concentration of the control is the same as in the control of the non-diluted medium condition and decreased by 5 times after overnight incubation with 10 μM erastin in comparison to the control (Figure 2c), and the protein concentration remained constant 450 $\pm$ 20 μg/mL for both conditions (Figure 2d). In both culture media, non-diluted, and diluted 2-times, erastin was able to inhibit system $X_c^-$, reducing the accessibility of cells to cystine/cysteine needed for GSH synthesis. Additionally, the high concentrations of cystine and glutamate in the culture medium, which were expected to compete with erastin on the binding on system $X_c^-$, did not affect erastin activity. Moreover, the 2 times diluted media was not deleterious for AoSMC viability after overnight incubation even though the cell culture components were reduced in 2 times diluted culture media.

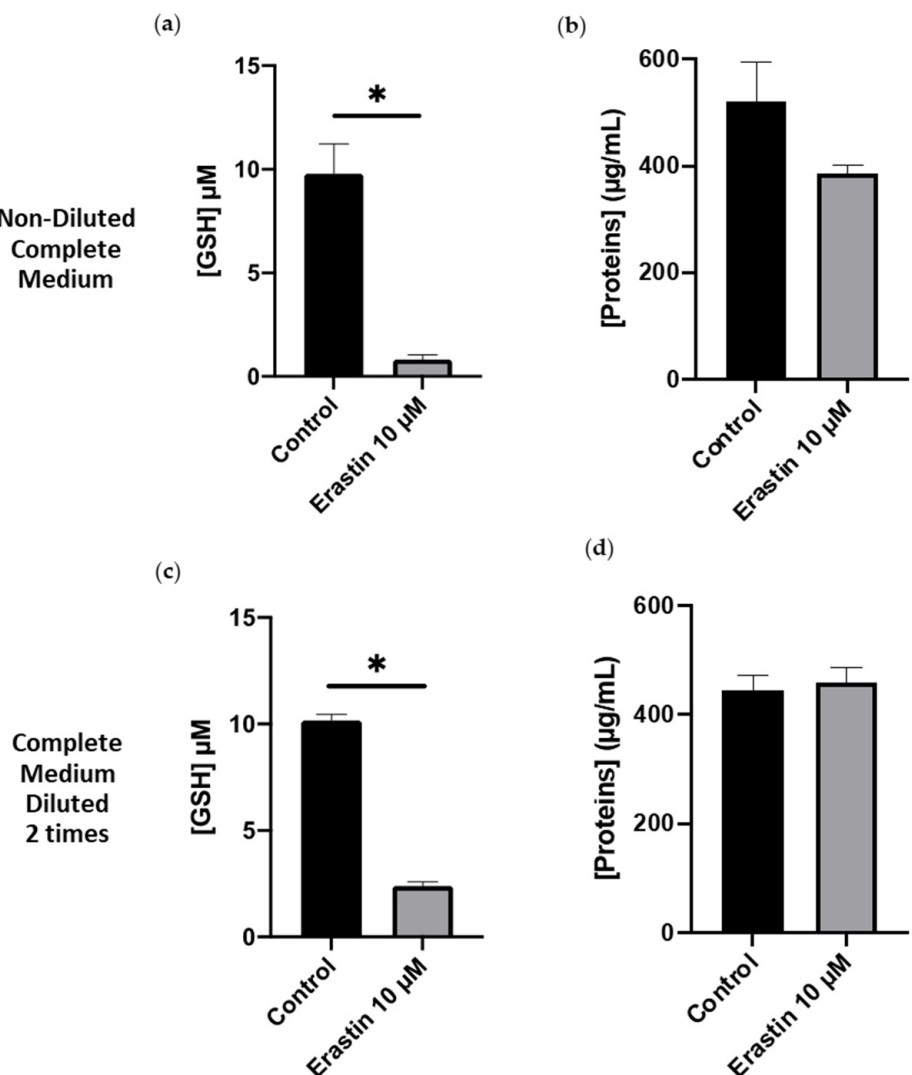

**Figure 2.** GSH and protein concentrations in AoSMCs after an overnight incubation with and without erastin in non-diluted complete medium (**a**,**b**) and complete medium diluted 2 times (**c**,**d**). Results are presented as mean ± sem of *n* = 3 experiments and compared using one sample *t*-test; * *p* < 0.05.

*3.3. Erastin-Induced Lipid Peroxidation in Different Dilution of Culture Media*

MDA was quantified to evaluate lipid peroxidation—the second ferroptosis biomarker—state of cells after overnight incubation with 10 μM erastin in non-diluted and in 2 times diluted complete media. In non-diluted complete medium, MDA concentration (Figure 3a) and protein concentration (Figure 3b) were determined at 0.1 ± 0.02 μM and 470 ± 50 μg/mL, respectively, and remained constant whatever the investigated conditions were (control or in the presence of erastin). In 2 times-diluted medium, MDA concentration fluctuated between 0.08 ± 0.05 μM in the control condition and 0.16 ± 0.05 μM in the erastin condition, without a significant difference between both conditions (Figure 3c). Protein concentration in both media (complete or 2 times diluted) was approximately 400 ± 80 μg/mL (Figure 3d). So, the blocking of the system $X_c^-$ with erastin is not sufficient to provoke lipid peroxidation in AoSMCs in both media after overnight incubation. This result was expected as lipid peroxidation is induced by hydroxyl radicals, produced upon the Haber–Weiss reaction resulting from iron ions oxidation. Therefore, in order to induce lipid peroxidation, we decided to introduce 50 μM of FC during erastin incubation, as a provider for ferric ($Fe^{3+}$) ions. The stability of FC in the present experimental conditions (aerobic conditions), makes it a compatible choice for the purpose of this study.

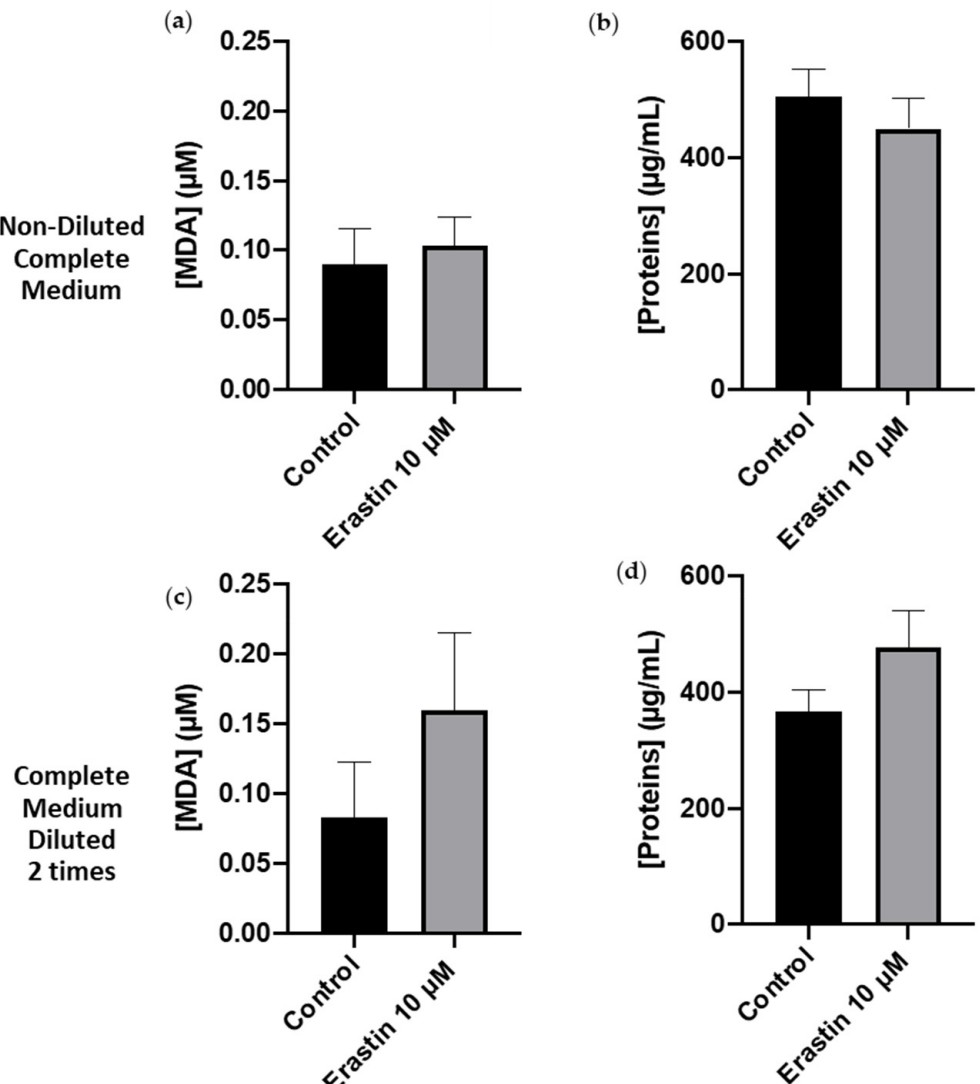

**Figure 3.** MDA and protein quantifications in AoSMCs after an overnight incubation with and without erastin in non-diluted complete medium (**a**,**b**) and complete medium diluted 2 times (**c**,**d**). Results are presented as mean $\pm$ sem of $n$ = 3 and compared using one-sample $t$-test.

### 3.4. Erastin-Induced Ferroptosis upon Iron Addition in Different Dilution of Culture Media

MDA concentration in non-diluted complete medium after overnight incubation of AoSMCs with 10 µM erastin + 50 µM FC did not significantly change compared both to the control and the 10 µM erastin conditions (Figure 4a). However, the protein concentration decreased by 3 times in the 10 µM erastin + 50 µM compared to the control and the 10 µM erastin conditions (Figure 4b). This means that, in complete medium, the introduction of 50 µM FC is deleterious to cells. Note that neither the chemical form nor the initial concentration of iron ions in the complete medium could not be communicated by the medium supplier. In complete medium diluted 2 times, the 10 µM erastin + 50 µM FC condition increased MDA concentration by 3.75 compared to the control condition (Figure 4c). In the meantime, the protein concentration stayed constant at approximately 420 $\pm$ 70 µg/mL in the 3 conditions (Figure 4d). This implies that, in complete medium diluted 2 times, 10 µM erastin + 50 µM FC promote lipid peroxidation without leading to cell death. It is worth noting that 50 µM FC alone did not have any impact on GSH or MDA concentrations.

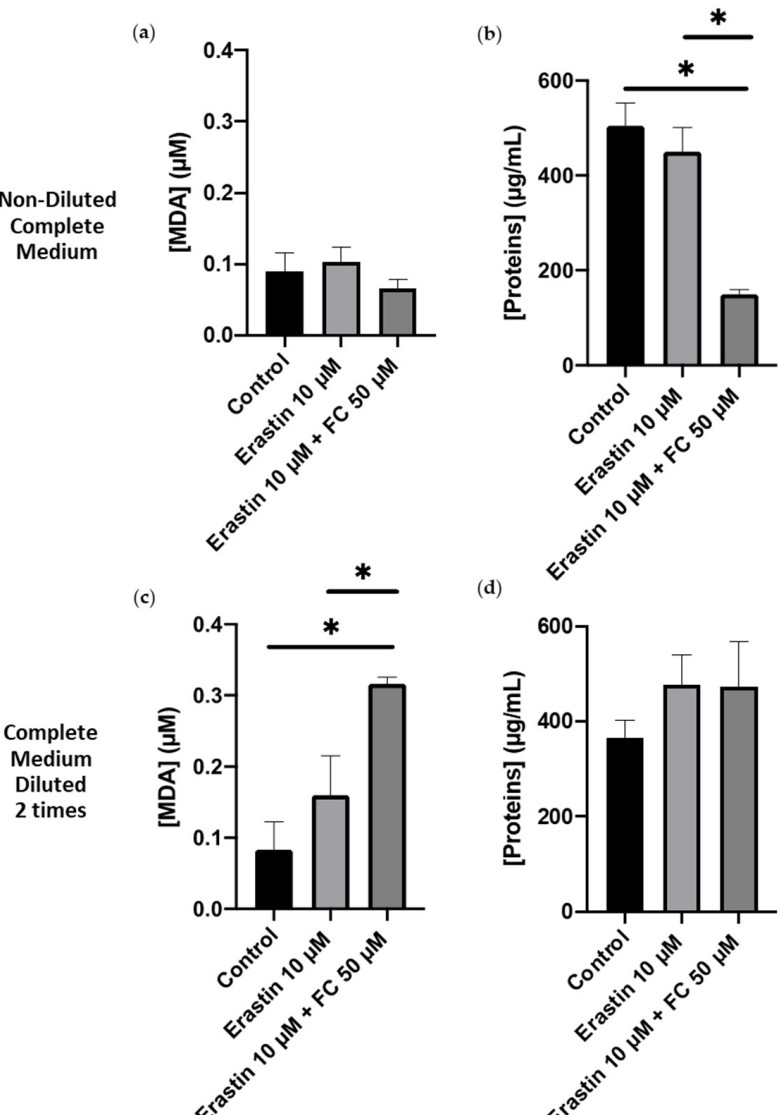

**Figure 4.** MDA and protein quantifications in AoSMC after an overnight incubation with and without erastin ± 50 μM FC in non-diluted complete medium (**a**,**b**) and in complete medium diluted 2 times (**c**,**d**). Results are presented as mean ± sem of *n* = 3 experiments and compared using one-way ANOVA and Tukey's multiple comparisons test * *p* < 0.05.

Therefore, inducing ferroptosis was validated using GSH and MDA quantifications upon overnight incubation with 10 μM Erastin (to act on cysteine supply and GSH synthesis) + 50 μM FC (to induce lipid peroxidation) in complete medium diluted 2 times.

To finally validate this ferroptosis model to evaluate iron-chelating molecules/peptides, we rescued this ferroptosis model using deferoxamine (DFO) and deferiprone (DFr), which are well known for their iron-chelating properties.

### 3.5. Rescue of the Erastin/FC-Induced Ferroptosis Model by Iron-Chelating Molecules

In complete medium diluted 2 times, intracellular GSH concentration was evaluated by plotting the ratio of GSH quantity (nmol) over protein mass (mg) (Figure 5a). The results show that this ratio is non-significantly increased by 50 μM DFO in comparison to the incubation with 10 μM erastin + 50 μM FC and is equivalent to the GSH ratio in the 10 μM erastin condition. However, 150 μM DFr is able to increase the GSH ratio compared to the 10 μM Erastin + 50 μM FC condition, but not significantly compared to the control condition (ratio = 30 ± 1 nmol/mg).

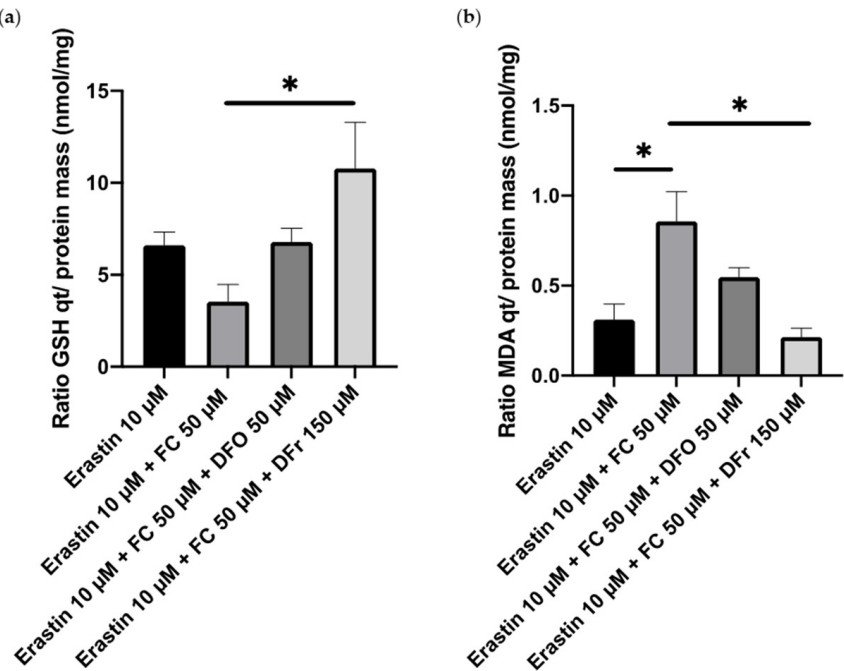

**Figure 5.** Ratio of GSH quantity over protein mass (**a**) and the ratio of MDA quantity over protein mass (**b**). Smooth muscle cells were incubated in complete medium diluted 2 times for 1 night with and without erastin ± 50 μM FC ± 50 μM deferoxamine (or 150 μM deferiprone). Results are presented as mean ± sem of *n* = 3 and compared using one-way ANOVA; * $p < 0.05$ (Tukey's multiple comparisons test).

Similarly, MDA concentration was evaluated by also plotting MDA quantity (nmol) over protein mass (mg) (Figure 5b). MDA concentration was non-significantly decreased after incubation with 50 μM DFO but significantly decreased by 4 times after incubation with 150 μM DFr in comparison to the incubation with 10 μM erastin + 50 μM FC.

## 4. Discussion

Ferroptosis is a form of regulated cell death characterized by iron-dependent accumulation of lipid peroxides [13]. The imbalance between pro-oxidants (such as lipid peroxides, reactive oxygen species, and free iron ions) and antioxidants (including glutathione and glutathione peroxidase 4) is crucial in regulating ferroptosis [15].

Despite ferroptosis being recognized as a cell death pathway involved in cardiovascular diseases like atherosclerosis, there is a scarcity of literature reporting treatments that specifically inhibit ferroptosis [15,16]. Furthermore, as far as the author is aware, there is a lack of a ferroptosis model based on cardiovascular cells, hindering the ability to test molecules for their potential to inhibit ferroptosis and, consequently, their applicability in the treatment of cardiovascular diseases. Our study involved the development of a ferroptosis model and its rescue in aortic smooth muscle cells (AoSMCs).

In the induction of ferroptosis, blocking system $X_c^-$ is a well-established mechanism [26]. Various strategies have been documented in the literature for this purpose, including the promotion of glutamate excess, modulation of the glutaminolysis pathway [27], and direct inhibition of system $X_c^-$ using erastin [28]. Erastin, a recognized inhibitor of system $X_c^-$, acts by reducing the intracellular concentration of cystine and, thus, impacting the synthesis of GSH [17]. Initially recognized as a cancer cell killer through a non-apoptotic pathway, erastin was later identified as an inducer of ferroptosis [13]. In this study, the comprehensive evaluation of erastin concentrations revealed a high degree of cytocompatibility with AoSMCs, as evidenced by the cells maintaining their metabolic activity across the tested range. The choice of 10 μM as the erastin concentration for subsequent experiments is

a judicious selection, ensuring both compatibility with cell viability and alignment with established practices in the field [20–22].

Quantification of intracellular GSH concentration serves as a crucial biomarker in the study of ferroptosis [17]. GSH, a tripeptide composed of glutamate, cysteine, and glycine, plays a pivotal role in cellular antioxidant defense mechanisms and redox homeostasis [14]. Monitoring changes in GSH levels provides valuable insights into the oxidative stress associated with ferroptosis. In our study, the observed decrease in intracellular GSH levels supports the hypothesis that erastin hinders the accessibility of cells to cystine/cysteine, essential for GSH synthesis. Furthermore, the experiment addressed concerns about the potential interference of high concentrations of cystine and glutamate in the culture medium with erastin's activity. Contrary to expectations, erastin's inhibitory effect on system $X_c^-$ remained evident, indicating its resilience to competition with these amino acids. Notably, the 2-times diluted culture medium did not compromise AoSMC viability after overnight incubation, emphasizing the robustness of the cells even under reduced culture components.

Quantification of GSH is often complemented by the assessment of lipid peroxidation markers (e.g., malondialdehyde) to provide a comprehensive understanding of ferroptotic processes. Therefore, MDA quantification was undertaken to assess the state of cells following overnight incubation with 10 μM erastin in non-diluted and 2-times diluted culture medium. The results indicate that erastin alone, even with the blockade of system $X_c^-$, is not adequate to induce lipid peroxidation in AoSMCs. This observation aligns with the understanding that lipid peroxidation is often induced by hydroxyl radicals, generated through the Haber–Weiss reaction resulting from the oxidation of iron ions [11].

To address this, ferric citrate (FC) was introduced during erastin incubation to provide ferric ($Fe^{3+}$) ions. Ferric iron is known for its involvement in redox reactions and its capacity to readily accept electrons, thereby becoming reduced to ferrous iron ($Fe^{2+}$) [11]. Several studies have reported the ferroptosis-inducing potential of iron supplementation through the introduction of ferric-based compounds, such as FC and ferric ammonium citrate, at varying concentrations [13,29,30].

In the context of complete medium, the introduction of 50 μM FC is detrimental to cell viability. The absence of information regarding the chemical form and initial concentration of iron ions in the complete medium further complicates the interpretation of these results. It is suspected that by adding 50 μM FC, the total concentration of iron ions for an overnight incubation is more than the cells can handle, causing oxidative damage and ultimately cell death. Therefore, the results validate the induction of ferroptosis in AoSMCs through the co-incubation of 10 μM erastin and 50 μM FC only in 2-times diluted complete medium. The observed increase in MDA concentration signifies the initiation of lipid peroxidation, a characteristic feature of ferroptosis. The sustained protein concentration indicates that, under these conditions, the cells undergo lipid peroxidation without undergoing significant cell death.

The development of a rescue model in AoSMC involving iron-chelating molecules, deferoxamine (DFO) and deferiprone (DFr), provides crucial positive control for future evaluation of other metal-chelating agents [17]. DFO, known for its effective iron chelation on a 1:1 basis [31], displayed some protective effect, albeit to a lesser extent. In contrast, DFr, with its capacity to bind $Fe^{3+}$ in a 3:1 molar ratio [32], demonstrated a more robust rescue effect. DFr was able to significantly increase GSH concentration and decrease MDA concentration. The small structure of DFr also facilitates its cell permeability, enabling it to effectively chelate intracellular iron. By chelating intracellular iron, deferiprone helps prevent iron from participating in harmful reactions that can lead to oxidative stress and tissue damage [33]. It also enables the excretion of excess iron from the body through urine, ultimately reducing the iron burden and mitigating the complications associated with iron overload disorders [31]. The observed rescue effect underscores the importance of targeting intracellular iron levels as a strategy to counteract ferroptosis.

## 5. Conclusions

In conclusion, the ferroptosis model is validated using GSH and MDA as biomarkers upon the overnight incubation of AoSMCs with erastin 10 μM and FC 50 μM. Erastin induced GSH depletion while the combination of erastin and ferric citrate—containing ferric ions ($Fe^{3+}$)—induced lipid peroxidation. After, DFO and DFr were investigated for their ability to rescue the ferroptosis cellular model, considering their well-known chelating properties. DFO (50 μM) was not effective on AoSMCs for rescuing ferroptosis, unlike DFr (150 μM).

It would be interesting to study the effect of synthetic metal-chelating peptides such as HHH, HHHHHH, HGH, and carnosine in inhibiting ferroptosis since it could be the start of a protein-based therapy (or prevention) for iron-overload diseases and then to investigate the effect of metal-chelating peptides discovered from protein hydrolysates [17].

Additionally, it is important to note that ferroptosis is a complex process influenced by various factors, including the activity of specific enzymes like glutathione peroxidase-4 and the presence of antioxidants [17]. Therefore, addressing ferroptosis may require a multifaceted approach that includes not only iron chelation but also the modulation of other cellular processes involved in lipid peroxidation and cell death. Researchers continue to study ferroptosis and develop new strategies and compounds to effectively inhibit this form of cell death in different pathological conditions.

**Author Contributions:** Conceptualization, C.G.; methodology, S.E.H., I.F.-R. and C.G.; validation, L.C.-R. and C.G.; formal analysis, S.E.H. and C.G.; writing—original draft preparation, S.E.H.; writing—review and editing, L.C.-R. and C.G.; supervision, C.G. All authors have read and agreed to the published version of the manuscript.

**Funding:** This research was funded by [French National Research Agency-ANR] grant number [15-004] and ANR JCJC MELISSA (2020).

**Institutional Review Board Statement:** Not applicable.

**Informed Consent Statement:** Not applicable.

**Data Availability Statement:** Data is contained within the article.

**Acknowledgments:** The authors acknowledge financial support from the "Impact Biomolecules" project of the "Lorraine Université d'Excellence" (in the context of the "Investissements d'avenir" program implemented by the French National Research Agency—ANR project number 15-004). The authors would like also to thank the financial support of ANR JCJC MELISSA (2020).

**Conflicts of Interest:** The authors declare no conflicts of interest.

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
