# Peer review of "A Smooth Muscle Cell-Based Ferroptosis Model to Evaluate Iron-Chelating Molecules for Cardiovascular Disease Treatment"

_cimb, doi:10.3390/cimb46020086_

Round 1

Reviewer 1 Report

Comments and Suggestions for Authors

Ferroptosis is implicated in the development of several diseases including cardiovascular diseases. The consequences of ferroptosis is the iron-dependent accumulation of oxidized phospholipids which leads to the programmed cell death. In this manuscript, El Hajj et al developed a model of ferroptosis on cardiovascular cells in order to test molecules able to inhibit ferroptosis with potential application in cardiovascular diseases treatment. For this, they used smooth muscle cells derived from human aorta and induced ferroptosis by erastin, the inhibitor of cystine/glutamate antiporter, and triggered lipid peroxidation with Fe(III) overload. As inhibitors of ferroptosis, iron chelating deferoxamine or deferiprone were used. Based on their experiments, they concluded that deferoxamine was not effective on AoSMCs for rescuing ferroptosis which was prevented by deferiprone. The manuscript is clearly written, however the experimental part of the manuscript has serious flaws and need numerous additional experiments.

1.      All experiments were done only with n = 3. This is not acceptable. Cell culture experiments are not expensive and time consuming and number of experiments should be at least 5 – 6 not less.

2.      Statistic. The parameters of statistical evaluations are not correct. The authors even did not tested whether their data are normally distributed and they can not use the tests which they used, therefore all conclusions based on statistical differences are not acceptable.

3.      In the methods it is not indicated whether the experiments were done in confluent cells, or in growing cells.

4.      P. 3, L. 112. It is not clear what means “pure triton”? % of triton (probably it was triton X-100) should be indicated.

5.      P. 4, L. 146. What the authors mean under “The intracellular protein concentration” After the cells are lysis, they can measure all protein content, including membrane proteins, not only intracellular proteins.

6.      Fig. 1. Control should be shown on this figure. 20% differences on cell viability (MTT test) is relatively high and it is not clear whether these differences are significant.

7.      P. 5 3.2. Erastin-induced intracellular GSH depletion in different dilution of culture media. It is absolutely not clear the reason why the authors performed such experiments. If they want to decrease cysteine/cystine and/or glutamate concentration in the media, they should ask the company to prepare such specific media but dilution of the media (especially 10 times) on which all other components of media (including FCS) will be diluted and make some conclusions from such kind of experiments is not acceptable.

8.      In the Fig. 2 three times indicated “non-diluted complete medium”, whereas on the figure legend (a & b), complete medium diluted 2 times (c & d) and complete medium diluted 10 times (e & f).  The same for Fig. 3. In Fig. 3 “t test; *p < 0.05”, however * in is not shown at Fig. 3. It should be corrected.

9.      Fig. 4a. Protein concentration reduced 3 times in non-diluted media after incubation with erastin + 50 μM ferric citrate, whereas in 2 times diluted media (Fig. 4b) it is not changed. How the authors can explain these very strange results? Does it mean that these conditions are toxic for cells? Cell viability (MTT test) should be included here and it should be recalculated as MDA/ total protein, not MDA/µM as it is calculated at Fig. 5.

In conclusions: all experiments should be repeated at least 5 – 6 times and appropriate statistic should be used. In the graphs better to indicate individual values. Experiments with diluted media are not informative, because concentrations of all components of the media will be diluted. If the authors want to decrease cysteine/cystine and/or glutamate concentration in the media they should ask company to prepare such specific media.

Author Response

Dear Reviewer

Thank you for your valuable feedback and constructive comments on our manuscript. We appreciate the opportunity to address your concerns regarding the experimental part of our manuscript.

  1. All experiments were done only with n = 3. This is not acceptable. Cell culture experiments are not expensive and time consuming and number of experiments should be at least 5 – 6 not less.

We acknowledge your point and would like to clarify that our study of a sample size n=3 was done in triplicates. We added a sentence in line 159 to clarify this “each experiment was done in triplicate”

  1. Statistic. The parameters of statistical evaluations are not correct. The authors even did not tested whether their data are normally distributed and they can not use the tests which they used, therefore all conclusions based on statistical differences are not acceptable.

We tested the normal distribution of or data using Shapiro-Wilks test before running further tests. We apologize for not mentioning it. Added on line 163 : “Normality was analyzed by Shapiro-Wilks test”

  1. In the methods it is not indicated whether the experiments were done in confluent cells, or in growing cells.

We added in line 89: “All experiments were conducted using actively growing cells at around 90% of confluency.

  1. P. 3, L. 112. It is not clear what means “pure triton”? % of triton (probably it was triton X-100) should be indicated.

Corrected in Line 115: pure Triton X-100

  1. P. 4, L. 146. What the authors mean under “The intracellular protein concentration” After the cells are lysis, they can measure all protein content, including membrane proteins, not only intracellular proteins.

In our context, "intracellular protein concentration" refers to the total protein content within the cellular compartment following cell lysis. We acknowledge your point that this includes not only soluble cytoplasmic proteins but also membrane proteins and other cellular components. We did not intend to exclude membrane proteins from our measurements. We changed this term to “cellular protein concentration”.

  1. Fig. 1. Control should be shown on this figure. 20% differences on cell viability (MTT test) is relatively high and it is not clear whether these differences are significant.

Control has been added to the figure. Statistical analysis confirmed no difference between control and erastin conditions.

  1. P. 5 3.2. Erastin-induced intracellular GSH depletion in different dilution of culture media. It is absolutely not clear the reason why the authors performed such experiments. If they want to decrease cysteine/cystine and/or glutamate concentration in the media, they should ask the company to prepare such specific media but dilution of the media (especially 10 times) on which all other components of media (including FCS) will be diluted and make some conclusions from such kind of experiments is not acceptable.

The decision to use diluted media was made with the intention of adjusting cystine/cysteine and glutamate concentrations, and that’s why in our primary experimental design we decided to test 3 conditions of the medium. Diluted 10 times, diluted 2 times and not diluted at all. You are totally right saying that it will modify the concentration of growth factors, FCS and antibiotics without affecting pH and osmolarity (PBS was used to dilute). This drastic modification brings us to the decision to finally exclude 10 times in the following experiments (lipid peroxidation evaluation and rescue model). We wanted to keep those results in the article in respect to the experimental design we followed. If you think it is not suitable, we can delete the results of 10 times dilution from this article.

We understand your suggestion to work with the media supplier to prepare specific media with controlled concentrations of cysteine/cystine and glutamate. However, it has become apparent that obtaining such custom media with the desired nutrient concentrations is not feasible at this time due to financial and technical problems.

  1. In the Fig. 2 three times indicated “non-diluted complete medium”, whereas on the figure legend (a & b), complete medium diluted 2 times (c & d) and complete medium diluted 10 times (e & f).  The same for Fig. 3. In Fig. 3 “t test; *p < 0.05”, however * in is not shown at Fig. 3. It should be corrected.

Figure caption have been corrected

  1. Fig. 4a. Protein concentration reduced 3 times in non-diluted media after incubation with erastin + 50 μM ferric citrate, whereas in 2 times diluted media (Fig. 4b) it is not changed. How the authors can explain these very strange results? Does it mean that these conditions are toxic for cells? Cell viability (MTT test) should be included here and it should be recalculated as MDA/ total protein, not MDA/µM as it is calculated at Fig. 5.

We agree with the strangeness of the results. The only explanation we found is that the basal culture medium already contains iron (concentration and chemical form not communicated by the supplier). Moreover, FCS contain also an unknowm concentration of iron. Therefore in non-diluted medium, the addition of iron overpassed the maximum concentration of iron tolerated by cells. Explanation have been clarified on lines 267-270

Reviewer 2 Report

Comments and Suggestions for Authors

The manuscript entitled A Smooth Muscle Cell-based Ferroptosis Model to Evaluate Iron Chelating Molecules for Cardiovascular Diseases Treatment is an original article. The authors aimed to develop a cell-based ferroptosis model upon the inhibition of the system Xc- antiporter and triggering lipid peroxidation with Fe(III)  overload. They validated this model using Fe(III) chelators such as deferoxamine and deferiprone as ferroptosis rescue molecules.

The manuscript is well written and well structured. The methods are very well detailed.

There are many acronyms; therefore, I suggest making a list of acronyms.

I recommend making, also, a figure with the role of ferroptosis, which is a complex process influenced by various factors. This could constitute a part of discussions. Otherwise, is very strange this manuscript without the chapter entitled discussions.

Author Response

Dear Reviewer,

Thank you for your thoughtful review and constructive feedback on our manuscript.

We appreciate your positive comments regarding the originality, writing, and structure of our manuscript. Your suggestion to create a list of acronyms has been duly noted, and we will make the necessary additions to enhance clarity for our readers. We first deleted all acronyms used only one to three times such as RDA, GPx4, RSL3, TMP, BCA, BSA.

Regarding your recommendation for a figure illustrating the role of ferroptosis, we would like to inform you that a similar figure has been previously included in one of our publications: El Hajj, S.; Canabady-Rochelle, L.; Gaucher, C. Nature-Inspired Bioactive Compounds: A Promising Approach for Ferrop-tosis-Linked Human Diseases? Molecules 2023, 28, 2636. However, to address your concern, we will consider incorporating a reference (14) to the relevant figure in our previous work, ensuring readers can access a detailed visual representation of the ferroptosis process.

In response to your suggestion about the absence of a dedicated "Discussions" section, we would like to clarify that we intentionally chose to integrate the results and discussion into a single section for a more cohesive narrative. This decision aligns with our approach to present findings and their interpretation simultaneously and is compatible with the journal recommendations.

Reviewer 3 Report

Comments and Suggestions for Authors

Authors used an experimental study to examine effect of erastin on GSH and MDA. However, this article has not fully answered some of the questions due to insufficient description and inadequate statistical analyses.

First, authors used ANOVA and t-test for comparison of groups, which had only 3 samples each other, but it may be inadequate statistical analyses due to small sample size. Authors should add discussion for justification or limitation in discussion section.

Second, authors mixed result section and discussion section, but findings and interpretation should be described separately as scientific article. Authors should rewrite result section and discussion section.

Third, authors concluded “Therefore, 10 µM of erastin is the first chosen concentration for the development of the ferroptosis model on AoSMC in our experiments.” (L173), but do not explain the criteria of “the first chosen concentration” in method section (e.g., the presence of significant statistical difference). Authors should add explanation in method section.

Fourth. Authors showed results in complete medium diluted 10 times for GSH in figure 2, but do not show those for MDA in figure 3 and figure 4. It is difficult to understand why authors use different method for MDA without explanation. Authors should add explanation in method section.

Finally, authors described some of sentences without citation or justification as follows; “Iron, an essential mineral in the human body, plays a crucial role in various physiological processes, such as hemoglobin production, oxygen transport and enzyme function.” (L27), “As free iron might be toxic due to its redox potential” (L29), “the body stores excess iron in the liver, where it can be mobilized when needed.” (L29), “Maintaining iron homeostasis is critically essential for human health to prevent diseases such as iron deficiency anemia and hemochromatosis caused by iron overload.” (L31), “Concentration of intracellular iron ions is regulated at several cellular levels that comprise: iron (Fe3+) uptake via transferrin receptors (TFR1 and TFR2), iron (Fe2+) storage in ferritin, and iron (Fe2+) export via ferroportin.” (L33), “Ferroptosis is a programmed cell death implicated in the development of several diseases such as hepatic cancer, renal and cardiovascular diseases (heart failure, atherosclerosis).” (L46), “The hallmark feature of ferroptosis is the iron-dependent accumulation of oxidized phospholipids (i.e., lipid peroxides).” (L48), “Ferroptosis can be “mimiced”/induced by the inactivation of system Xc (cystine/glutamate antiporter) using either erastin, sorafenib, sulfasalazine, or excess glutamate or also, by inhibiting the glutathione peroxidase 4 (GPx4) using Ras Selective Lethal 3 (RSL3).” (L49), “The inhibition of the system Xc-induces the depletion of intracellular cysteine pool, the limiting amino acid required for glutathione (GSH) synthesis.” (L52), “the decrease of intracellular GSH concentration causes an imbalance in cell redox homeostasis and antioxidant capacity.” (L54), “it inhibits the glutathione-dependent enzymes, such as the GPx4 that catalyzes the reduction of lipid peroxides, while oxidizing GSH.” (L55), “ferroptosis has been identified as a cell death pathway occurring in cardiovascular diseases such as atherosclerosis” (L61), “very few studies in the literature report treatments inhibiting ferroptosis.” (L62), “the Beer Lambert equation.” (L143), “Tukey’s post-test.” (L158), “By chelating intracellular iron, deferiprone helps prevent iron from participating in harmful reactions that can lead to oxidative stress and tissue damage.” (L295), and “Additionally, it is important to note that ferroptosis is a complex process influenced by various factors, including the activity of specific enzymes like GPx4 and the presence of antioxidants.” (L318), but it is difficult for readers to judge it without references as evidence for each description. Authors should add references for these descriptions.

Minor comments

Figure 2. “Non-Diluted Complete Medium” in (c) and (d) may be “complete medium diluted 2 times”.

Figure 2. “Non-Diluted Complete Medium” in (e) and (f) may be “complete medium diluted 10 times”.

Figure 3. “Non-Diluted Complete Medium” in (c) and (d) may be “complete medium diluted 2 times”.

Figure 4. “Non-Diluted Complete Medium” in (c) and (d) may be “complete medium diluted 2 times”.

All of figures. The lower limits of interval are not shown in each bar.

Author Response

Dear Reviewer

Thank you for your valuable feedback and constructive comments on our manuscript. We appreciate the opportunity to address your concerns

First, authors used ANOVA and t-test for comparison of groups, which had only 3 samples each other, but it may be inadequate statistical analyses due to small sample size. Authors should add discussion for justification or limitation in discussion section.

We acknowledge your point that a higher number of experiments would provide more robust results. However, we would like to clarify that our study of a sample size n=3 was done in triplicates. We added a sentence in line 159 to clarify this “each experiment was done in triplicate”. We also tested the normal distribution of or data using Shapiro-Wilks test before running further tests. We apologize for not mentioning it. Added on line 163: “Normality was analyzed by Shapiro-Wilks test”

Second, authors mixed result section and discussion section, but findings and interpretation should be described separately as scientific article. Authors should rewrite result section and discussion section.

We would like to clarify that we intentionally chose to integrate the results and discussion into a single section for a more cohesive narrative. This decision aligns with our approach to present findings and their interpretation simultaneously. According to the journal’s guidelines, this representation is accepted.

Third, authors concluded “Therefore, 10 µM of erastin is the first chosen concentration for the development of the ferroptosis model on AoSMC in our experiments.” (L173), but do not explain the criteria of “the first chosen concentration” in method section (e.g., the presence of significant statistical difference). Authors should add explanation in method section.

We added a clearer explanation in line 175-179: “According to some studies, 10 µM of erastin is the commonly used concentration to inhibit the system Xc- and induce ferroptosis [11–13]. Therefore, a concentration of 10 µM for erastin was chosen in order to insure the inhibition of system Xc- necessary for the development of the ferroptosis model on AoSMC in our experiments.”

Fourth. Authors showed results in complete medium diluted 10 times for GSH in figure 2, but do not show those for MDA in figure 3 and figure 4. It is difficult to understand why authors use different method for MDA without explanation. Authors should add explanation in method section.

We would like to clarify that the decision to not experiment in complete medium diluted 10 times for MDA was based on the preliminary findings from our initial experiments with GSH and protein quantification (Figure 2). The 10 times dilution of complete medium promoted a too drastic decrease in GSH and protein concentrations compared to non-diluted complete medium. This might be deleterious to cells and prompted us not to continue with this condition. We added an elucidated statement on Line 235-236

Finally, authors described some of sentences without citation or justification as follows ... but it is difficult for readers to judge it without references as evidence for each description. Authors should add references for these descriptions.

In the revised version of the manuscript, we thoroughly reviewed the identified sentences and ensure that each statement is appropriately cited.

Minor comments

Figures: The figures issues have been addressed

All of figures. The lower limits of interval are not shown in each bar.

To avoid any figure overloading, we decided to not add the lower bar as it is commonly accepted.

Round 2

Reviewer 1 Report

Comments and Suggestions for Authors

The authors adequately addressed most points. I will suggest to delete the results of 10 times dilution of media from this article.

Concentration of “pure Triton X-100” should be indicated.

Fig. 4. In the first round I asked that “Cell viability (MTT test) should be included here and it should be recalculated as MDA/ total protein, not MDA/µM”, however the authors did not followed these suggestions. I will strongly recommend to include MTT test and recalculate the data.

Author Response

The authors adequately addressed most points. I will suggest to delete the results of 10 times dilution of media from this article.

The 10-times dilution have been removed

Concentration of “pure Triton X-100” should be indicated.

The concentration has been indicated: “10% (w/v)” Line 117

Fig. 4. In the first round I asked that “Cell viability (MTT test) should be included here and it should be recalculated as MDA/ total protein, not MDA/µM”, however the authors did not follow these suggestions. I will strongly recommend to include MTT test and recalculate the data.

The calculated MDA/total protein is represented in Fig 5 just like GSH/protein only for the 2-times diluted medium condition. Calculating the MDA/total protein for non-diluted medium condition is not scientifically relevant as protein concentration is here to normalize our compared results. The high decrease of protein concentration will show an erroneous variation of MDA that will only depend on the variation of protein concentration linked to the disappearance of cell thus their death. In our opinion, no need of MTT test to assume that the huge decrease in protein concentration is correlated with the death of adherent cells.

Reviewer 3 Report

Comments and Suggestions for Authors

Authors revised the manuscript, but this article has not fully answered some of the questions due to insufficient description and inadequate statistical analyses.

First, as mentioned in the first review, authors used ANOVA and t-test for comparison of groups, which had only 3 samples each other, but it may be inadequate statistical analyses due to small sample size. Authors suggest “we would like to clarify that our study of a sample size n=3 was done in triplicates”, but statistical methods remain inadequate. Moreover, authors suggest “Normality was analyzed by Shapiro-Wilks test”, but the results are not shown in the manuscript.

Second, as mentioned in the first review, authors mixed result section and discussion section, but findings and interpretation should be described separately as scientific article. Authors suggest “We would like to clarify that we intentionally chose to integrate the results and discussion into a single section for a more cohesive narrative. This decision aligns with our approach to present findings and their interpretation simultaneously. According to the journal’s guidelines, this representation is accepted.”, but the mixture of result and discussion section makes understanding this manuscript difficult. Moreover, authors revised following sentences; “According to some studies, 10 µM of erastin is the commonly used concentration to inhibit the system Xc- and induce ferroptosis [20–22]. Therefore, a concentration of 10 µM for erastin was chosen in order to insure the inhibition of system Xc- necessary for the development of the ferroptosis model on AoSMC in our experiments.” (L179), but if so, these explanations should be shown in method section. Furthermore, authors added the following sentence; “Note that the deleterious effects of diluting the culture medium by 10 times prompted us not to continue with this condition.” (L239) and suggest “We would like to clarify that the decision to not experiment in complete medium diluted 10 times for MDA was based on the preliminary findings from our initial experiments with GSH and protein quantification (Figure 2). The 10 times dilution of complete medium promoted a too drastic decrease in GSH and protein concentrations compared to non-diluted complete medium. This might be deleterious to cells and prompted us not to continue with this condition.”, but if so, this should be included in the methods as an explanation of the overall design of the study. The mixture of result and discussion section makes understanding this manuscript difficult.

Third, as mentioned in the first review, the lower limits of interval in all of figures are not shown in each bar. Authors suggest “To avoid any figure overloading, we decided to not add the lower bar as it is commonly accepted.”, but it is difficult for readers to understand what authors did without details of results.

Finally, as mentioned in the first review, authors described some of sentences without citation or justification. Authors suggest “In the revised version of the manuscript, we thoroughly reviewed the identified sentences and ensure that each statement is appropriately cited.”, but some of them remain without references as follows; “Ferroptosis is a programmed cell death implicated in the development of several diseases such as hepatic cancer, renal and cardiovascular diseases (heart failure, atherosclerosis).” (L47), “The hallmark feature of ferroptosis is the iron-dependent accumulation of oxidized phospholipids (i.e., lipid peroxides). Ferroptosis can be “mimiced”/induced by the inactivation of system Xc-(cystine/glutamate antiporter) using either erastin, sorafenib, sulfasalazine, or excess glutamate or also, by inhibiting the glutathione peroxidase 4 using Ras Selective Lethal 3.” (L49), “The inhibition of the system Xc-induces the depletion of intracellular cysteine pool, the limiting amino acid required for glutathione (GSH) synthesis.” (L53), and “Yet, the decrease of intracellular GSH concentration causes an imbalance in cell redox homeostasis and antioxidant capacity.” (L55). but it is difficult for readers to judge it without references as evidence for each description.

Author Response

Authors revised the manuscript, but this article has not fully answered some of the questions due to insufficient description and inadequate statistical analyses.

First, as mentioned in the first review, authors used ANOVA and t-test for comparison of groups, which had only 3 samples each other, but it may be inadequate statistical analyses due to small sample size. Authors suggest “we would like to clarify that our study of a sample size n=3 was done in triplicates”, but statistical methods remain inadequate. Moreover, authors suggest “Normality was analyzed by Shapiro-Wilks test”, but the results are not shown in the manuscript.

Due to limited laboratory resources and funding constraints, we carefully optimized our experimental design to balance the need for robust results with the practical limitations we faced. To limit the number of samples to n=3 in triplicate, we used the Shapiro-Wilks test part to assess normality. You are totally right; the result of the Shapiro-Wilks test should be mentioned in the manuscript. Therefore, we clarified the result of the test by adding the sentence: “All data were analyzed by Shapiro-Wilks test and showed p-value > 0.05, suggesting that there is no significance that the data deviates from a normal distribution” Line 163-165

Second, as mentioned in the first review, authors mixed result section and discussion section, but findings and interpretation should be described separately as scientific article. Authors suggest “We would like to clarify that we intentionally chose to integrate the results and discussion into a single section for a more cohesive narrative. This decision aligns with our approach to present findings and their interpretation simultaneously. According to the journal’s guidelines, this representation is accepted.”, but the mixture of result and discussion section makes understanding this manuscript difficult.

We have separated the results and discussion sections

Moreover, authors revised following sentences; “According to some studies, 10 µM of erastin is the commonly used concentration to inhibit the system Xc- and induce ferroptosis [20–22]. Therefore, a concentration of 10 µM for erastin was chosen in order to insure the inhibition of system Xc- necessary for the development of the ferroptosis model on AoSMC in our experiments.” (L179), but if so, these explanations should be shown in method section.

We have moved this explanation to the material and method, section chapter 2.4. Induction and rescue of ferroptosis, Line 111-115, as proposed

Furthermore, authors added the following sentence; “Note that the deleterious effects of diluting the culture medium by 10 times prompted us not to continue with this condition.” (L239) and suggest “We would like to clarify that the decision to not experiment in complete medium diluted 10 times for MDA was based on the preliminary findings from our initial experiments with GSH and protein quantification (Figure 2). The 10 times dilution of complete medium promoted a too drastic decrease in GSH and protein concentrations compared to non-diluted complete medium. This might be deleterious to cells and prompted us not to continue with this condition.”, but if so, this should be included in the methods as an explanation of the overall design of the study. The mixture of result and discussion section makes understanding this manuscript difficult.

The 10 times dilution was removed according to the suggestion of reviewer 1

Third, as mentioned in the first review, the lower limits of interval in all of figures are not shown in each bar. Authors suggest “To avoid any figure overloading, we decided to not add the lower bar as it is commonly accepted.”, but it is difficult for readers to understand what authors did without details of results.

Our decision to not include the lower limits of intervals in each bar was made to avoid figure overloading, aligning with common practices in the field. It is true that in many cell culture studies, including the lower limits is not a widespread practice. As the lower limit is equal to the upper limit, no detail of the results was lost. In response to your feedback and to enhance clarity for readers we added in the method section: “The lower limit of SEM interval in equal to the upper one” Line 162-163

Finally, as mentioned in the first review, authors described some of sentences without citation or justification. Authors suggest “In the revised version of the manuscript, we thoroughly reviewed the identified sentences and ensure that each statement is appropriately cited.”, but some of them remain without references as follows; “Ferroptosis is a programmed cell death implicated in the development of several diseases such as hepatic cancer, renal and cardiovascular diseases (heart failure, atherosclerosis).” (L47), “The hallmark feature of ferroptosis is the iron-dependent accumulation of oxidized phospholipids (i.e., lipid peroxides). Ferroptosis can be “mimiced”/induced by the inactivation of system Xc-(cystine/glutamate antiporter) using either erastin, sorafenib, sulfasalazine, or excess glutamate or also, by inhibiting the glutathione peroxidase 4 using Ras Selective Lethal 3.” (L49), “The inhibition of the system Xc-induces the depletion of intracellular cysteine pool, the limiting amino acid required for glutathione (GSH) synthesis.” (L53), and “Yet, the decrease of intracellular GSH concentration causes an imbalance in cell redox homeostasis and antioxidant capacity.” (L55). but it is difficult for readers to judge it without references as evidence for each description.

We apologize for these omissions. The issue of the references has been considered according to your feedback

Round 3

Reviewer 3 Report

Comments and Suggestions for Authors

Authors revised the manuscript, but this article has not fully answered some of the questions due to insufficient description.

First, as mentioned in the first and second review, authors used ANOVA and t-test for comparison of groups, which had only 3 samples each other, but it may be inadequate statistical analyses due to small sample size. Authors suggest “Due to limited laboratory resources and funding constraints, we carefully optimized our experimental design to balance the need for robust results with the practical limitations we faced.”, but if so, authors should add further description regarding small sample size issues and its uncertainty to the limitation section.

Second, as mentioned in the first the second review, authors mixed result section and discussion section, and authors added discussion section (L298). However, the heading of “Results and Discussion” remains (L172). Authors should revise the manuscript, carefully.

Minor comments

L159. “acid  assay” may be “acid assay”.

Comments on the Quality of English Language

Minor comments

L159. “acid  assay” may be “acid assay”.

Author Response

Authors revised the manuscript, but this article has not fully answered some of the questions due to insufficient description.

First, as mentioned in the first and second review, authors used ANOVA and t-test for comparison of groups, which had only 3 samples each other, but it may be inadequate statistical analyses due to small sample size. Authors suggest “Due to limited laboratory resources and funding constraints, we carefully optimized our experimental design to balance the need for robust results with the practical limitations we faced.”, but if so, authors should add further description regarding small sample size issues and its uncertainty to the limitation section.

We would like to insist on the fact that it is not “only 3 samples” but n=3 in triplicate. Therefore, we can clarify this as follows “3 different wells (triplicate) per group from 3 independent experiments in each group (n=3)”. This statement has been added in the statistical analysis section following by “The number of samples has been reduced to the smallest size allowing to show statistical differences between conditions”. Indeed, after checking the normal distribution of results, the graph pad prism software allowed us to analyse those results using ANOVA and t-test.

Second, as mentioned in the first the second review, authors mixed result section and discussion section, and authors added discussion section (L298). However, the heading of “Results and Discussion” remains (L172). Authors should revise the manuscript, carefully.

The heading “Results and Discussion” has been corrected for “Results”

Minor comments

L159. “acid  assay” may be “acid assay”.

The space has been deleted

Round 4

Reviewer 3 Report

Comments and Suggestions for Authors

Authors revised the manuscript, and I have no further comments.